# Innovation and Utilization of Functional Feed Additives from Maize By-Products in Broiler Chickens

**DOI:** 10.3390/ani14223198

**Published:** 2024-11-07

**Authors:** Orranee Srinual, Chanidapha Kanmanee, Phatchari Srinual, Thanongsak Chaiyaso, Mongkol Yachai, Tanya Tapingkae, Wanaporn Tapingkae

**Affiliations:** 1Department of Animal and Aquatic Sciences, Faculty of Agriculture, Chiang Mai University, Chiang Mai 50200, Thailand; orranee.s@cmu.ac.th (O.S.); chanidapha.k@cmu.ac.th (C.K.); phatchari_sri@cmu.ac.th (P.S.); 2Functional Feed Innovative Center (FuncFeed), Faculty of Agriculture, Chiang Mai University, Chiang Mai 50200, Thailand; 3Office of Research Administration, Chiang Mai University, Chiang Mai 50200, Thailand; 4Division of Biotechnology, Faculty of Agro-Industry, Chiang Mai University, Chiang Mai 50100, Thailand; thanongsak.c@cmu.ac.th; 5Faculty of Animal Science and Technology, Maejo University, Chiang Mai 50290, Thailand; mongkol_yc@mju.ac.th; 6Department of Technology and Agricultural Development, Faculty of Agricultural Technology, Chiang Mai Rajabhat University, Chiang Mai 50330, Thailand; tanya_tap@cmru.ac.th

**Keywords:** spent mushroom substrate, enzyme, antibiotic, performance, broilers

## Abstract

This study explored the effects of incorporating spent mushroom substrate (SMS) from *Flammulina velutipes* as an exogenous enzyme in broiler diets. The objectives of this study were to evaluate the impact of being fed varying levels of SMS on the growth performance, carcass characteristics, meat quality, blood chemistry, cecal microbiota, and small intestinal morphology of broilers. The findings indicate that dietary supplementation with SMS significantly enhances growth performance, improves carcass characteristics, reduces serum cholesterol, and positively influences gut health.

## 1. Introduction

The development and growth of the livestock feed industry has resulted in an ever-increasing demand for corn, causing a boom in Thailand’s mountainous northern region [1]. Following corn harvesting, corncobs represent a by-product that can be employed in various applications to enhance livestock feed [2] and produce biofuels [3]. Furthermore, corncobs have emerged as a promising and sustainable substrate for mushroom cultivation [4].

Corncobs are nutrient-rich, providing essential components for the growth of fungal mycelium, including carbohydrates, cellulose, and lignin [5,6]. The residue generated from mushroom cultivation, referred to as spent mushroom substrate (SMS), can be effectively repurposed as a valuable resource in animal feed and holds the potential for enzyme production [6].

Research has shown that spent mushroom substrate (SMS) can enhance animal feed quality. Specifically, incorporating SMS from *Cordyceps militaris* improved growth performance, immunoglobulin secretion, antioxidant capacity, and cholesterol levels in growing pigs [7]. Additionally, fermented SMS supplementation benefits the intestinal mucosal barrier, immunity, and microbiota composition [8]. In ruminants, SMS from *Pleurotus sajorcaju* increases the degradable fiber fraction and improves feed intake [9]. Mushrooms are also noted for their high nutritional value and medicinal benefits [10]. Fard et al. [11] reported that mushrooms contain immune-boosting substances, including polysaccharides, glycosides, alkaloids, and selenium, which enhance immunity and combat free radicals in poultry. As a result, the use of mushrooms and their by-products as dietary supplements and antibiotic alternatives in broiler farming has attracted considerable attention in recent years [12].

*Flammulina velutipes* is among the most widely consumed edible mushrooms, recognized for its rich nutrient profile, which includes protein, dietary fiber, carbohydrates, and vitamins [13,14,15,16]. This species exhibits a range of bioactivities, such as antioxidant properties, immunomodulation, anti-inflammatory effects, hepatoprotection, antitumor activity, anti-hyperlipidemic effects, cognitive enhancement, and resistance to senescence [17,18]. However, limited research exists concerning the effects of spent mushroom substrate (SMS) from *F. velutipes* on broiler chickens. Therefore, the objective of this study was to assess the impact of dietary SMS from *F. velutipes* on growth performance, blood chemistry, carcass characteristics, meat quality, cecal microbial composition, and small intestinal morphology in broilers.

## 2. Materials and Methods

### 2.1. Animals and Experiment Design

The animals utilized in this study were handled in accordance with the IACUC guidelines, and study procedures were approved by the Institutional Animal Care and Use Committee of Chiang Mai University (AG02010/2566). A total of 500 ROSS 308 broilers (one day of age) were divided into 5 groups each comprising ten replicates (10 chicks per replicate). The five groups included a control diet (CON), AGP (which consisted of amoxicillin 100 g kg^−1^ and colistin 400 × 10^6^ IU kg^−1^ at 0.25 g kg^−1^; Otamix a.c., Octa Memorial Co., Ltd., Bangkok, Thailand), and spent mushroom substrate supplementation at 0.5 g kg^−1^ (SMS0.5), 1.0 g kg^−1^ (SMS1.0), and 2.0 g kg^−1^ (SMS2.0) of diet. Feeding and water were provided ad libitum for 35 days. The ingredients and chemical compositions of the diet are presented in Table 1. The enzymatic activities, including xylanase, endoglucanase, cellulase, and laccase, in SMS were assessed using the methods outlined by Tapingkae et al. [19]. Additionally, the chemical composition of SMS is detailed in Table 2, while the enzymatic activities are presented in Figure 1.

### 2.2. Growth Performance Measurement

The body weight of the broilers was recorded on day 1 of the experiment. Data was collected on the daily feed intake and once-a-week body weight, which were measured for each replicate (cage). Average daily gain (ADG), average daily feed intake (ADFI), feed conversion ratio (FCR), and feed cost per unit of gain were calculated and analyzed based on the weekly data collected. The mortality rate for each group was recorded throughout the trial.

### 2.3. Measurement of Blood Biochemicals

On day 35, one bird from each replicate was randomly selected for blood sampling from the wing vein. Serum was obtained by centrifugation at 3000× *g* per minute and 4 °C for 15 min. Then, the serum was stored at −20 °C for determination of the blood biochemical parameters [20] including total cholesterol, triglyceride, high-density lipoprotein cholesterol (HDL), and low-density lipoprotein cholesterol (LDL). Additionally, blood urea nitrogen (BUN), creatinine, aspartate transaminase (AST), alanine aminotransferase (ALT), alkaline phosphatase (ALP), total protein, albumin, and globulin were measured by Latimer [21].

### 2.4. Characterization of Carcass and Meat Quality

On day 35, following a 4 h fasting period, and in accordance with the average body weight, one broiler from each replicate was randomly selected from each treatment group. These broilers were then weighed and subsequently slaughtered. According to Underwood and Anthony [22], slaughtering chicks were euthanized for collection data. The chicks were weighed to calculate the live weight, defeather weight, carcass weight, carcass percentage, carcass composition, meat weight, and internal organs for carcass characteristics analysis. Furthermore, the breast and thigh were immediately packed individually in sealable plastic bags and stored at 4 °C for meat quality measurement [20], with the pH value measured at 15 min and 24 h using a portable pH meter. At 24 h, the breast meat was assessed for the color values (lightness; L*, redness; a*, and yellowness; b*) using a Minolta Chroma Meter, Model CR-410, Minolta Camera Co., Ltd., Osaka, Japan [23], and the drip loss value was measured by hanging 2 g of samples in a plastic bag, sealing them, and storing them at 4 °C. Shear force and cooking loss were determined following the method described by Qu et al. [24]. Samples of meat were prepared with a weight of 50–60 g. The meat sample was gently wiped with tissue paper to remove the surface water and the weight was recorded, and the sample was then placed in a heat-resistant bag and sealed securely, before being soaked in a heat bath at 80 °C for 20 min. Then, the sample was removed and allowed to cool at room temperature for 30–35 min. The surplus liquids were absorbed after taking out the sample from the bag and the final weight was recorded to calculate the cooking loss. For the shear force analysis, the cooked meat samples were cut into squares with a thickness of 1.27 cm and the cutting force was measured with a mechanical device (Instron Model 3433 Universal test machine, Norwood, MA, USA).

### 2.5. Composition of Microflora in Cecal Contents

Cecal digesta were collected at the end of the trial and counted for bacteria using the modified protocol of Qiu et al. [25]. One gram (1 g) of caeca digesta was collected and deposited in sterile sampling tubes with 9-mL of 0.1% peptone solution, which was then mixed using a vortex mixer. Then, a dilution of the cecal sample was plated onto *Lactobacillus* MRS Agar Granulated plates (Himedia Laboratories, Maharashtra, India), Xylose-Lysine Deoxycholate (XLD) Agar plates (Himedia Laboratories, Maharashtra, India), and EMB Agar Levine plates (Himedia Laboratories, Maharashtra, India) to isolate and culture *Lactobacilli* sp., *Salmonella* sp. and *Escherichia coli*, respectively. The XLD and EMB Agar plates were incubated at 37 °C for one day, while the Lactobacilli MRS Agar plates were incubated at 37 °C for two days before being removed and counted immediately. The count was measured as log_10_ of colony-forming units (cfu) per gramme of cecum.

### 2.6. Measurement of Intestinal Histomorphology

Following slaughter, 1 cm segments of the duodenum, jejunum, and ileum from the broiler chickens were rinsed with phosphate-buffered saline (PBS) and subsequently fixed in a 10% formaldehyde solution for 24 h. Subsequently, the tissue specimens underwent a standardized three-step processing protocol, dehydration, clearing, and impregnation with paraffin wax [26]. Using a rotary microtome (Medite model A550, Burgdorf, Germany), paraffin-embedded tissues were cut into 4 μm-thick sections and arranged on glass slides. Deparaffinized tissue specimens were stained with haematoxylin and eosin. After staining, intestinal tissue sections were mounted and coated with a glass coverslip. A compound microscope (A1 Zeiss Axio Scope, Carl Zeiss, Gottingen, Germany) was used to detect the histology sections at 10× magnification. After that, Motic Images Plus 2.0 software (Motic China Group Co., Xiamen, Fujian, China) was used to assess villus height (VH), villus width (VW), crypt depth (CD), muscularis mucosae thickness (MMT), and VH:CD ratio with an image analyzer [27].

### 2.7. Statistical Analysis

The recorded data were analyzed using the SPSS program version 23.0 (SPSS Inc., Chicago, IL, USA). The effect of spent mushroom substrate supplementation was examined using a one-way ANOVA. Duncan’s New Multiple Range Test was used to determine the differences between the groups. Differences within the *p*-value < 0.05 were considered significant. The results of the different levels of SMS supplementation were analyzed using linear and quadratic contrast.

## 3. Results

### 3.1. Growth Performance

The effect of dietary spent mushroom substrate (SMS) on the growth performance of broilers is shown in Table 3. Throughout the entire rearing period (days 1–35), broilers receiving the antibiotic (AGP) and SMS treatments demonstrated a significant increase in final body weight (FBW) compared to the basal diet group (CON) (*p* < 0.001). Notably, the AGP group exhibited the highest FBW among the treatments. Additionally, broilers fed an AGP diet showed significantly higher ADFI and ADG compared to the other groups (*p* < 0.001). Both the AGP and SMS2.0 groups demonstrated a reduced feed conversion ratio and lower feed cost per gain (*p* < 0.05). Mortality rates did not vary significantly among the different dietary groups. In addition, higher SMS supplementation tended to result in a linear improvement in FBW, ADFI, ADG, FCR, and FCG (*p* < 0.001), while only FBW, ADFI, and ADG showed a quadratic increase (*p* < 0.05).

### 3.2. Serum Biochemical

Supplementation with SMS1.0 and SMS2.0 significantly lowered the serum cholesterol content of broilers (Table 4). In contrast, broilers that were fed the CON and SMS0.5 diets exhibited elevated blood cholesterol levels (*p* < 0.001). Additionally, the inclusion of AGP in the diet significantly increased blood triglyceride levels in broilers (*p* < 0.001). Broilers on the CON diet showed a significant increase in HDL cholesterol levels (*p* < 0.05), although no significant difference was observed when compared to broilers fed the SMS0.5 diet. Furthermore, LDL cholesterol levels were elevated in broilers fed the CON diet (*p* < 0.05). The supplementation of SMS did not affect blood urea nitrogen, creatinine, aspartate transaminase, alanine aminotransferase, alkaline phosphatase, total protein, albumin, or globulin levels in comparison to the CON and AGP diet groups. Supplementation with SMS resulted in a linear decrease in serum cholesterol, excluding triglycerides (*p* < 0.05). Likewise, there was a quadratic improvement in cholesterol, LDL cholesterol, and blood urea nitrogen levels in broilers that received SMS (*p* < 0.05).

### 3.3. Carcass Characteristics and Meat Quality

Dietary supplementation with SMS significantly improved the live weight of broilers (*p* < 0.05), though this effect was not statistically different when compared to broilers fed the AGP diet, as detailed in Appendix A. Broilers on the SMS2.0 diet demonstrated an increased defeather weight, carcass weight, and carcass percentage (*p* < 0.05); however, this result did not differ significantly from those observed in the AGP and SMS 1.0 groups. In addition, live weight, defeather weight, and carcass weight increased linearly with higher levels of SMS supplementation (*p* < 0.05). Conversely, no significant differences were observed in carcass composition, meat percentage, or internal organ percentage between the dietary supplementation groups and the CON diet (*p* > 0.05). Notably, higher levels of SMS supplementation led to a linear reduction in wing and gizzard weight (*p* < 0.05) while also resulting in a quadratic increase in skeleton (*p* < 0.05) among broilers.

Appendix A presents the effects of varying levels of SMS supplementation on the meat quality of broilers. The results indicate that SMS supplementation did not significantly impact meat quality attributes, including pH values, color (both breast and thigh), drip loss, cooking loss, or shear force, compared to the CON and AGP diets. However, there was a notable linear and quadratic trend improvement in color and drip loss (*p* < 0.05) as the dietary concentration of SMS increased.

### 3.4. Ceacal Microbials

Supplementation with SMS in the broilers’ diets significantly reduced the presence of pathogenic microorganisms (*E. coli* and *Salmonella* sp.), while concurrently increasing the abundance of beneficial microorganisms (*Lactobacillus* sp.), in the cecum of the broilers, compared to those fed the control (CON) diet (*p* < 0.001). In contrast, dietary supplementation with AGP resulted in a significant reduction in both pathogenic and beneficial microorganisms (*p* < 0.001) relative to the other treatment groups, as illustrated in Table 5. Notably, supplementation with SMS led to a significant quadratic decrease in the count of *E. coli* and *Salmonella* sp. in the ceacal contents as dietary SMS levels increased (*p* < 0.05).

### 3.5. Small Intestine Histology

Supplementation with SMS significantly reduced crypt depth (CD) in the duodenum of broilers compared to those receiving the AGP and CON diets (*p* < 0.05), as detailed in Table 6. Additionally, broilers on the AGP diet demonstrated the highest villus height to crypt depth (VH:CD) ratio in the duodenum, though this ratio did not significantly differ from that observed in broilers receiving SMS1.0 and SMS2.0 diets. In the jejunum, broilers receiving the SMS1.0 or SMS2.0 diets exhibited a statistically significant increase in VH compared to those on the AGP diet (*p* < 0.05). Additionally, CD was highest in broilers fed the CON and SMS2.0 diets. Moreover, ileal morphology, including VH, VW, CD, MMT, and VH:CD, was not significantly influenced by SMS supplementation (*p* > 0.05). However, CD in the duodenum and jejunum and musculalis mucosea thickness in the jejunum were improved quadratically (*p* < 0.05) by increasing the concentration of SMS in the diets.

## 4. Discussion

Spent mushroom substrate (SMS) has received an outstanding amount of attention as a significant waste product, which is composed of residual fungal mycelium, lignocellulosic biomass, and enzymes [6,28,29]. The utilization of enzymes in poultry feed is already well known and has nutritional advantages that can be obtained without impacting the final product’s quality. Furthermore, there have been novel findings about the potential of products that provide efficient enzymes as additions to animal diets [30]. However, the composition of raw SMS varies according to the source of the mushroom medium used for cultivation. SMS is also a necessary source of vitamins and minerals, as well as being rich in bioactive compounds [31]. Incorporating SMS in animal feed has been used for several decades, and improves microbial balance and growth in livestock and poultry [6].

In this study, SMS supplementation linearly improved the final body weight (FBW), average daily gain (ADG), and feed conversion ratio (FCR) in broilers that were fed the control diet (CON), especially the SMS2.0 diet. This finding aligns with the research conducted by Machado et al. [32], which demonstrated that the inclusion of SMS derived from *Agaricus blazei* at a concentration of 0.2% in broiler diets led to improvements in weight gain and FCR, positioning SMS as a viable alternative to antibiotic additives. However, higher concentrations of SMS at 0.4% were found to adversely affect growth performance. Moreover, supplementation with SMS from *Pleurotus sajor-caju* did not impact feed intake or FCR; nevertheless, the addition of SMS at 0.67% during the initial 21 days resulted in increased body weight [33]. Aderemi et al. [34] reported that replacing 25% of wheat with SMS from *Pleurotus ostreatus* in starter broiler diets significantly improved body weight gain (BWG) and FCR. Conversely, finisher broilers exhibited lower BWG compared to other groups, alongside an increased FCR. Additionally, SMS and by-products from mushroom cultivation have been incorporated into poultry feed. Fermented substrates from *Pleurotus eryngii* [35] or *Hypsizygus marmoreus* [36] can be utilized in layer feed at concentrations ranging from 5% to 15% without detrimental effects on performance. Furthermore, the inclusion of oyster mushroom stems at a level of 30 g kg^−1^ was shown to improve body weight and FCR in quails, while feed intake remained unaffected [37]. In swine nutrition, the supplementation of SMS from *Cordyceps militaris* at 2 g kg^−1^ significantly increased FBW without affecting average daily feed intake (ADFI) or FCR [7]. Liu et al. [38] observed that the addition of stem waste from *Flammulina velutipes* in pig feed at elevated levels resulted in reduced ADFI while improving the G:F ratio. The by-products of mushroom cultivation manifest in various forms and species. Chang et al. [39] reported that replacing 5% of the diet with spent mushroom compost in geese did not negatively impact growth performance and could lower production costs. Furthermore, Parichaya et al. [40] determined that supplementation with *Cordyceps militaris* substrate at 2.5 g kg^−1^ was optimal for broiler diets. SMS is rich in bioactive components, including polysaccharides, vitamins, and trace elements, which help improve the balance of digestive tract flora in poultry and promote their growth [41]. These discussions stem from various factors, including the mushroom species, dosage levels, application methods (fermented or non-fermented and combined with other beneficial organisms), the specific mushroom parts used (such as fruiting bodies or stem bases), and the treatment duration [42]. However, there is broad consensus among scientists that mushrooms can positively impact the performance and overall health of broiler chickens [12,43,44]. The potential mechanisms associated with mushrooms include alterations in intestinal microbiota, enhanced nutrient digestibility, increased feed intake and absorption, improved nitrogen absorption, a strengthened immune response, and enhanced antioxidant activity [45].

The current analysis of the lipid profile in broilers demonstrated a linear reduction in cholesterol, triglycerides, high-density lipoprotein (HDL), and low-density lipoprotein (LDL) cholesterol levels in birds fed a diet supplemented with spent mushroom substrate (SMS). The concentration of triglycerides and cholesterol in the blood is a crucial indicator of lipid metabolism, reflecting adipose tissue development and the extent of fat deposition [12]. Our findings are consistent with those of Mahfuz et al. [46], who reported that serum total cholesterol and high-density lipoprotein cholesterol levels decreased linearly with the inclusion of *F. velutipes* mushroom stem waste in broiler chickens. The observed reduction may be attributed to the extract’s ability to inhibit the 3-hydroxy-3-methylglutaryl coenzyme A (HMG-CoA) reductase activity in the liver, the enzyme responsible for cholesterol synthesis, particularly at higher administration levels. This effect is likely due to lovastatin, a known inhibitor of HMG-CoA reductase, which disrupts mevalonate production. The concentration of lovastatin is sufficient to bind to HMG-CoA reductase, preventing the formation of mevalonic acid—a key compound in cholesterol biosynthesis—thereby inhibiting cholesterol formation [47]. Beta-glucan and its derivatives in medicinal mushrooms have cholesterol-lowering effects by reducing absorption or increasing fecal excretion [48]. Feeding mushrooms may also suppress endogenous cholesterol biosynthesis by inhibiting the activity of HMG-CoA reductase, the rate-limiting enzyme in cholesterol synthesis [49]. This finding is consistent with research conducted by Mahfuz et al. [50] which demonstrated that the inclusion of mushroom stems from *Flammulina velutipes* at concentrations of 1% and 2% in broiler diets significantly decreased cholesterol and HDL levels compared to a control diet. Additionally, Parichaya et al. [40] reported that supplementation with mushroom stem waste from *Cordyceps militaris* at an optimal level of 2.5 g kg^−1^ did not adversely affect blood metabolites, while also contributing to a reduction in cholesterol and LDL cholesterol levels.

Furthermore, the addition of oyster mushroom extract to broiler diets did not significantly influence total protein, albumin, globulin, cholesterol, or triglyceride levels [47]. Lima et al. [51] indicated that the use of *Agaricus subrufescens* and *Pleurotus ostreatus* mushrooms did not significantly impact protein and albumin levels in the blood of 21-day-old broilers compared to control groups. However, in 42-day-old broilers, those receiving diets containing antibiotic growth promoters and *Pleurotus ostreatus* exhibited reductions in protein and albumin levels, respectively. Additionally, supplementation with *Agaricus subrufescens* and *Pleurotus ostreatus* resulted in decreased triglyceride levels, although cholesterol levels remained unaffected. In related studies, Nasir et al. [37] found that dietary supplementation with oyster mushroom stems did not influence levels of blood urea nitrogen, creatinine, aspartate transaminase, alanine aminotransferase, or alkaline phosphatase in quails. It was observed that feeding with different levels of SMS had no significant impact on liver and kidney functions. The values of ALT, AST, ALP, blood urea, and creatinine remained consistent across all groups, indicating no discernible differences in these parameters. The results of our study suggest that the improved lipid profiles associated with SMS supplementation may reduce the burden of protein synthesis in the liver and regulate fat deposition, thereby benefiting the cardiovascular system. In broilers supplemented with spent mushroom substrate (SMS), neither carcass characteristics nor meat quality demonstrated significant improvement. This finding is corroborated by Azevedo et al. [33], who reported that the incorporation of SMS from *Pleurotus sajor-caju* at levels of 0.5%, 1.0%, 1.5%, and 2.0% did not affect carcass quality. Similarly, Machado et al. [36] indicated that the use of SMS derived from *Agaricus blazei* in broiler diets did not impact carcass yield. Aderemi et al. [34] found that replacing wheat bran with SMS from *Pleurotus ostreatus* did not significantly alter the weights of various broiler parts, including breast, thighs, back, neck, wings, and shoulders. Meat quality can be described based on its physical and nutritional attributes. The physical properties are influenced by genetic lineage and nutrition [52]. The authors concluded that mushrooms can be incorporated into chicken feed without affecting meat quality [53].

The observed quadratic changes in the cecal microbial composition in the group treated with spent mushroom substrate (SMS) are consistent with findings from Chaiwat et al. [54] who reported an enhancement of Lactobacillus and Bifidobacterium in the ceca of layer hens fed a diet supplemented with SMS derived from *Cordyceps militaris*. This supplementation also resulted in a reduction of *Clostridium* species, coliforms, and *Escherichia coli*. In a related study, Lee et al. [55] reported a significant linear decrease in *Salmonella* sp. numbers in the cecum, dropping from 5.036 to 3.031 log_10_ CFU/g, when *Flammulina velutipes* mushrooms were included in the feed at levels of 0–50 g/kg. Additionally, *E. coli* numbers showed a significant quadratic reduction, decreasing from 5.405 to 4.759 log_10_ CFU/g. Mushrooms contain polysaccharides composed of l-arabinose, d-galactose, d-galacturonic acid, and d-glucuronic acid in a ratio of 18:18:1:1. These polysaccharides have biological effects that may enhance gut microbiota, leading to the production of fermentation products like volatile fatty acids (VFA), which could contribute to an increase in cell populations in the gastrointestinal tract (GIT) [56]. SMS includes fiber, which can promote intestinal health by regulating the composition and metabolism of bacterial communities [57]. Furthermore, Liu et al. [38] demonstrated that the inclusion of 2.5% waste from golden needle mushroom stems in pig diets enhanced the population of lactic acid bacteria in the gut and improved overall gut health. These strong antimicrobial effects of mushrooms might be due to the ability to change cell membranes and cause ion leakage, thus, making microbes less virulent [58].

The inclusion of dietary spent mushroom substrate (SMS) significantly influenced the morphology of the duodenum and jejunum in broilers. Greater intestinal villus height indicates an enhanced capacity for nutrient absorption in the intestine [59]. Additionally, higher villi are associated with active cell mitosis, which provides greater absorptive potential for various nutrients [60]. Deeper intestinal crypts suggest rapid tissue metabolism to facilitate the renewal of intestinal villi [61]. Moreover, a higher ratio of villus height to crypt depth results in lower maintenance requirements, leading to improved growth efficiency due to reduced turnover of the intestinal mucosa [62]. Research conducted by Azevedo et al. [33] indicated that higher levels of SMS derived from *Pleurotus ostreatus* (ranging from 0% to 2%) resulted in a reduction of villus height (VH) in the small intestine of broilers. In contrast, Liu et al. [38] found that incorporating waste from golden needle mushroom stems at a concentration of 2.5% in pig diets enhanced gut health by improving the villus height/crypt depth ratio (VH:CD) in the small intestines, thereby stimulating microbial diversity and increasing populations of beneficial bacteria. Furthermore, the addition of *Agaricus bisporus* stalk meal at a level of 20 g/kg in feed was observed to improve the morphology of the small intestine in broilers, evidenced by an increase in both the VH:CD in the ileum and the VH in the jejunum and ileum [63]. Lima et al. [51] also reported that the dietary supplementation of oyster and shiitake mushrooms in starter broilers led to an increase in villus width (VW) in the duodenum and crypt depth (CD) in the jejunum. However, no significant differences were observed when compared to the antibiotic growth promoter (AGP) group. Notably, VH was greater in 42-day-old broilers fed a control diet compared to those supplemented with oyster mushrooms and AGP. The enhancement of gut morphology through mushroom supplementation may be attributed to their antioxidant properties. The observed improvement in the intestinal health of chickens is attributed to the presence of biologically active compounds such as fiber, glucans, phenols, and other active ingredients in mushrooms [10,64]. Ye et al. [57] indicated that dietary fiber provides energy to colonic epithelial cells, promotes the production of intestinal mucosa, stimulates intestinal motility, and helps maintain intestinal integrity [65]. In addition, gut microbiota play a crucial role in improving gut morphology in poultry, with a positive and direct correlation identified between villus height and the abundance of gut microbes [66,67,68].

## 5. Conclusions

This study found that spent mushroom substrate (SMS) from *Flammulina velutipes* improves growth performance, reduces serum cholesterol, and positively affects the composition of cecal microbiota in broilers. While SMS led to significant weight gains and better feed conversion ratios, these were not as pronounced as those with antibiotic growth promoters (AGPs). SMS also promoted beneficial gut bacteria and reduced pathogenic strains without negatively impacting meat quality. Histological analysis showed improved intestinal health with reduced crypt depth and increased villus height. Overall, SMS is a promising alternative to AGPs in broiler diets, supporting growth and gut health.

## Figures and Tables

**Figure 1 animals-14-03198-f001:**
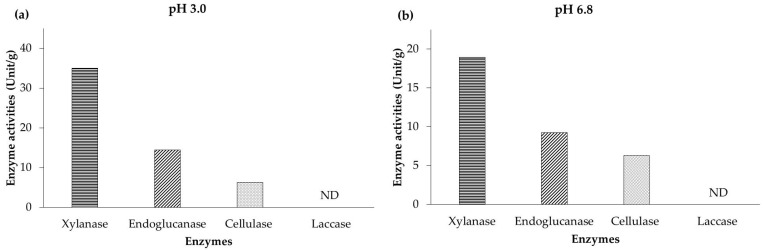
The enzyme activities of spent mushroom substrate at pH 3.0 (**a**) and 6.8 (**b**), respectively.

**Table 1 animals-14-03198-t001:** The experimental diet’s ingredients and chemical composition.

Item	Starter (1–14 d)	Finisher (15–35 d)
Ingredients (%)		
Yellow corn	57.17	59.58
Raw rice bran	5.00	5.00
Soybean meal (48% CP)	28.62	20.74
Rapeseed meal (38% CP)	3.20	6.00
Pork meal (50% CP)	3.00	4.00
Soybean oil	-	2.94
Salt	0.26	0.14
L-lysine	0.10	0.08
DL-methionine	0.25	0.15
Biofoss (21% P) ^1^	0.70	0.02
Calcium carbonate	1.07	0.64
Premixes ^2^	0.63	0.71
Proximate analysis (% as fed)		
Dry matter	89.53	89.54
Ash	5.92	5.92
Crude protein	21.02	19.60
Crude fiber	5.83	6.11
Crude fat	4.06	4.35
Gross energy	3882.03	3961.17

^1^ Biofoss supplies per kg of diet: monocalcium phosphate 1000 g, calcium 150 g, phosphorus 210 g, fluorine 0.21%. ^2^ Mineral premix supplies per kg of diet: vitamin A 20,000,000 IU, vitamin D3 4,000,000 IU, vitamin E 11,000 IU, vitamin K3 4.00 g, vitamin B1 5.00 g, vitamin B2 10.00 g, vitamin B6 5.00 g, vitamin B12 0.02 g, vitamin C 15.00 g, pantothenic acid 15.00 g, folic acid 3.00 g, nicotinic acid 40.00 g, biotin 0.20 g, magnesium 100.00 g, potassium 90.00 g, sodium 100.00 g, and feed additive 25.30 g.

**Table 2 animals-14-03198-t002:** The chemical composition of spent mushroom substrate (SMS) from *Flammulina velutipes* (% as fed).

Item	SMS
Dry matter	94.88
Moisture	5.12
Ash	10.51
Crude protein	8.73
Ether extract	3.71
Crude fiber	22.41
Nitrogen-Free Extract	54.64

**Table 3 animals-14-03198-t003:** The growth performance in broilers that were fed varying levels of spent mushroom substrate.

Item	CON	AGP ^1^	SMS0.5	SMS1.0	SMS2.0	SEM	*p*-Value		
							T	L	Q
Initial BW (g)	41.97	42.17	43.67	42.27	43.07	0.232	0.146	-	-
Final BW (g)	1930.20 ^c^	2254.80 ^a^	2113.80 ^b^	2143.90 ^b^	2196.60 ^a,b^	20.389	<0.001	<0.001	<0.05
ADFI (g/d)	78.54 ^c^	84.75 ^a^	84.52 ^a^	84.20 ^a,b^	83.78 ^b^	0.345	<0.001	<0.001	<0.001
ADG (g/d)	53.95 ^c^	63.22 ^a^	59.15 ^b^	60.05 ^b^	61.53 ^a,b^	0.583	<0.001	<0.001	<0.05
FCR	1.46 ^a^	1.34 ^b^	1.43 ^a^	1.40 ^a,b^	1.37 ^b^	0.011	0.002	<0.05	0.683
FCG (baht/kg)	57.31 ^a^	52.98 ^b^	56.10 ^a^	55.09 ^a,b^	53.48 ^b^	0.415	0.002	<0.05	0.722

^1^ Antibiotic (amoxicillin and colistin at 0.25 g kg^−1^); SMS0.5, spent mushroom substrate addition at 0.5 g kg^−1^; SMS1.0, spent mushroom substrate addition at 1.0 g kg^−1^; SMS2.0, spent mushroom substrate addition at 2.0 g kg^−1^. Body weight (BW); average daily feed intake (ADFI); average daily gain (ADG); feed conversion ratio (FCR); feed cost per gain (FCG). ^a–c^ Mean values in the same row with different letters indicate significant differences (*p*-value < 0.05). SEM, standard error of the mean. T is the *p*-value of the effect of treatment on analyzed traits; L and Q are the *p*-values of the linear and quadratic contrasts of SMS supplementation.

**Table 4 animals-14-03198-t004:** The lipid profile and blood chemistry in broilers that were fed varying levels of spent mushroom substrate.

Item	CON	AGP ^1^	SMS0.5	SMS1.0	SMS2.0	SEM	*p*-Value		
							T	L	Q
Lipid profiles (mg/dL)									
Cholesterol	140.80 ^a^	136.00 ^b^	140.80 ^a^	130.40 ^c^	134.60 ^b^	0.994	<0.001	0.013	0.033
Triglyceride	76.40 ^b^	105.00 ^a^	84.00 ^b^	77.80 ^b^	79.60 ^b^	2.458	<0.001	0.851	0.545
HDL-cholesterol	103.80 ^a^	97.40 ^b,c^	102.80 ^a,b^	94.00 ^c^	95.40 ^c^	1.135	0.006	<0.05	0.129
LDL-cholesterol	42.60 ^a^	33.60 ^b^	36.00 ^b^	33.20 ^b^	34.00 ^b^	1.020	0.007	<0.05	<0.05
Blood chemistry									
Blood urea nitrogen (mg/dL)	1.04	0.80	0.76	0.72	0.88	0.048	0.237	0.495	<0.05
Creatinine (mg/dL)	0.28	0.29	0.26	0.28	0.31	0.009	0.640	0.223	0.552
Aspartate transaminase (U/L)	281.40	307.20	298.80	336.40	313.20	9.498	0.492	0.314	0.241
Alanine aminotransferase (U/L)	2.80	2.20	3.00	3.40	2.60	0.173	0.261	0.712	0.198
Alkaline phosphatase (U/L)	2257.60	2272.80	2324.60	2252.20	2269.60	19.356	0.808	0.844	0.760
Total protein (mg/dL)	3.06	3.10	2.84	2.90	2.72	0.064	0.322	0.088	0.767
Albumin (mg/dL)	1.26	1.28	1.28	1.26	1.18	0.022	0.605	0.166	0.413
Globulin (mg/dL)	1.80	1.82	1.56	1.64	1.54	0.054	0.322	0.177	0.495

^1^ Antibiotic (amoxicillin and colistin at 0.25 g kg^−1^); SMS0.5, spent mushroom substrate addition at 0.5 g kg^−1^; SMS1.0, spent mushroom substrate addition at 1.0 g kg^−1^; SMS2.0, spent mushroom substrate addition at 2.0 g kg^−1^. HDL-cholesterol; high-density lipoprotein-cholesterol; LDL-cholesterol; low-density lipoprotein-cholesterol. ^a–c^ Mean values in the same row with different letters indicate significant differences (*p*-value < 0.05). SEM, standard error of the mean. T is the *p*-value of the effect of treatment on analyzed traits; L and Q are the *p*-values of the linear and quadratic contrasts of SMS supplementation.

**Table 5 animals-14-03198-t005:** The ceacal microbial content of broilers that were fed varying levels of spent mushroom substrate (log^10^ cfu/g).

Item	CON	AGP ^1^	SMS0.5	SMS1.0	SMS2.0	SEM	*p*-Value		
							T	L	Q
*E. coli*	7.51 ^c^	6.13 ^a^	6.40 ^b^	6.43 ^b^	6.56 ^b^	0.128	<0.001	0.052	<0.001
*Salmonella* sp.	6.09 ^c^	4.59 ^a^	5.08 ^b^	5.13 ^b^	5.34 ^b^	0.136	<0.001	0.123	<0.05
*Lactobacillus* sp.	6.82 ^c^	6.53 ^d^	7.29 ^b^	7.37 ^a,b^	7.45 ^a^	0.097	<0.001	0.166	0.282

^1^ Antibiotic (amoxicillin and colistin at 0.25 g kg^−1^); SMS0.5, spent mushroom substrate addition at 0.5 g kg^−1^; SMS1.0, spent mushroom substrate addition at 1.0 g kg^−1^; SMS2.0, spent mushroom substrate addition at 2.0 g kg^−1^. ^a–d^ Mean values in the same row with different letters indicate significant differences (*p*-value < 0.05). SEM, standard error of the mean. T is the *p*-value of the effect of treatment on analyzed traits; L and Q are the *p*-value of the linear and quadratic contrast of SMS supplementation.

**Table 6 animals-14-03198-t006:** The small intestinal histology in broilers that were fed varying levels of spent mushroom substrate.

Item	CON	AGP ^1^	SMS0.5	SMS1.0	SMS2.0	SEM	*p*-Value		
							T	L	Q
**Duodenum**									
Villus height (µm)	6217.84	6441.03	5278.11	5746.29	6061.53	181.541	0.302	0.859	0.228
Villus width (µm)	847.09	787.33	700.05	699.21	693.45	24.921	0.187	0.111	0.146
Crypt depth (µm)	1694.83 ^a^	1176.09 ^b^	1365.24 ^b^	1263.97 ^b^	1372.81 ^b^	54.251	0.017	0.106	<0.005
Muscularis mucosae thickness (µm)	120.31	160.79	193.13	167.01	174.52	9.194	0.136	0.242	0.156
Villus height/crypt depth ratio	3.69 ^b^	5.54 ^a^	4.04 ^b^	4.60 ^a,b^	4.46 ^a,b^	0.194	0.019	0.142	0.303
**Jejunum**									
Villus height (µm)	5262.35 ^a,b^	4715.43 ^b^	5365.10 ^a,b^	5563.42 ^a^	5965.14 ^a^	130.103	0.026	0.052	0.835
Villus width (µm)	607.95	639.16	592.92	650.33	554.70	20.222	0.618	0.446	0.315
Crypt depth (µm)	1008.87 ^a^	773.79 ^b^	718.17 ^b^	884.61 ^a,b^	981.48 ^a^	33.788	0.010	0.586	<0.05
Muscularis mucosae thickness (µm)	251.26	183.16	147.44	200.50	241.81	14.122	0.106	0.579	<0.05
Villus height/crypt depth ratio	5.25	6.28	7.49	6.63	6.21	0.276	0.136	0.679	0.063
**Ileum**									
Villus height (µm)	3656.09	3438.99	3833.73	3639.95	3901.41	102.079	0.764	0.548	0.828
Villus width (µm)	711.42	754.86	667.79	756.30	597.58	35.567	0.624	0.386	0.468
Crypt depth (µm)	740.75	771.30	715.28	682.79	725.01	25.276	0.879	0.866	0.535
Muscularis mucosae thickness (µm)	207.44	246.18	218.51	185.98	260.92	12.344	0.331	0.244	0.301
Villus height/crypt depth ratio	5.07	4.47	5.45	5.60	5.46	0.204	0.436	0.613	0.540

^1^ Antibiotic (amoxicillin and at 0.25 g kg^−1^); SMS0.5, spent mushroom substrate addition at 0.5 g kg^−1^; SMS1.0, spent mushroom substrate addition at 1.0 g kg^−1^; SMS2.0, spent mushroom substrate addition at 2.0 g kg^−1^. ^a,b^ Mean values in the same row with different letters indicate significant differences (*p*-value <0.05). SEM, standard error of the mean. T is the *p*-value of the effect of treatment on analyzed traits; L and Q are the *p*-values of the linear and quadratic contrasts of SMS supplementation.

## Data Availability

Most of the data generated or analyzed in this study are presented in this published article or its Supplementary Information. Additional data not included here are accessible upon reasonable request to the corresponding author.

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
