# Peer review of "Innovation and Utilization of Functional Feed Additives from Maize By-Products in Broiler Chickens"

_animals, 2024, doi:10.3390/ani14223198_

Round 1

Reviewer 1 Report

Comments and Suggestions for Authors

Title: Change title to “Innovation and utilization of Functional Feed Additives from Maize By-products in broiler chickens”

Abstract Section: Poorly written. Needs improvement. Contains more of introductory statements than the results of the study.

Line 20 -21: Rephrase to “The objectives of this study were to evaluate the impact of feeding varying levels of SMS on growth performance, carcass characteristics, meat quality, cecal microbiota, and small intestinal morphology in broilers.”

Line 34-35: Rephrase. First line of result should not read “however”. Was it significant? Statement should read “…broilers in CON group had decreased average daily feed intake compared to ……”. Include other important significant parameters….

Line 35-37: Check the conclusion part.

Line 41-43: Provide Reference

Line 51-55: while the emphasis on enzymes. Delete.

Line 57: change “improve to “improved”

Line 65: remove “medicinal”

Line 70-71: Provide more references.

Line 81: Remove “were”

Line 82: Use “Ross”

Line 84-85: Rewrite as “consisted of amoxicillin 100 g kg-1 and colistin 400×106 85 IU kg-1 at 0.25 g kg-1 ; Otamix a.c., Octa Memo-84 rial Co., Ltd., Bangkok, Thailand)

Line 95: Stick with one “Mortality or mortality rate”

Line 98: Change “broiler” to “bird”

Line 120-124: Authors should rephrase this section of the procedure to provide more clarity. Not clear “The meat sample was absorbed moisture…….

Line 127: Provide details of the “mechanical device”.

Line 131: Remove “briefly”. Should read One gram (1 g)

Line 132: Rephrase this “9-ml 0.1% peptone solution combined with a vortex mixer.”

Line 133: To what dilution factor? Provide the precise composition of the MRS and EMB plates.

Line 144: Rewrite as “Subsequently, the tissue specimens underwent a standardized three step processing protocol,”

Line 145: Remove “which included three primary steps:”

Line 146: Provide model of the rotary microtome

Line 155: Author should run linear regressions to model and analyse the relationships between the variables assessed and SMS supplementation

Line 168: Delete “In contrast, the inclusion of SMS at a concentration of 2.0 g/kg (SMS2.0) did not significantly affect ADFI (p > 0.05).” Not correct.

Line 167: Did authors use P<0.01???? Adjust this.

Line 173: Statement not correct. AGP, SMS1.0 and SMS2.0 had significantly lowered Serum Cholesterol content. Check your Table.

Line 191: Start on a new line or paragraph.

Line 197: Authors should note that healthy chickens have beneficial E. coli in their intestines. Is the E. coli in the study beneficial or pathogenic as assumed by the Authors?

Line 210-212: Delete the line.

Line 215-217 Delete.

Line 220: No details on Energy of diets in “Table 1. Ingredients and chemical composition of the experimental diets.”

Line 227-229: Inconsistency in assigning of superscripts. E.g. FCR, the lowest had the first alphabet unlike the other parameters

Line 254: Check values of the redness. It is suspect.

Line 227-280: Figure 1 does not show anything. Not necessary. Authors did not use any annotations.

Line 284-285: What is the link to the enzyme??

Line 297-318: Authors only cited works and did not provide scientific reasons/justifications for their results. Authors should review more on the composition of SMS and link its bioactive components in relation to some of the performance indices. Too much of unnecessary citations.

Line 320: What is the implication of the reduction in the HDL-Cholesterol? Authors failed to provide reasons for the reduction in the cholesterol composition of the serum. Authors can read and check: Ekunseitan, D. A., Ekunseitan, O. F., Odutayo, O. J., & Adeyemi, P. T. (2017). Pleurotus Ostreatus: Its Effect on Carcass, Serum Metabolites and Meat Lipoprotein Content of Broiler Chickens. Pertanika Journal of Tropical Agricultural Science, 40(4): 629 – 638 for help. There more examples.

Line 330: Authors should check Table 3. Delete “cholesterol, or triglyceride levels:

Line 342-345: Delete the line.

Line 353-362: Delete the lines.

Line 378-394: Too much citations. Authors has not properly explained the reasons for the changes in the morphological changes in the duodenum and the jejunum and it’s implication on the birds.

Line 394: How is this achieved? Explain this.

Results: Still more corrections

Discussion: Not properly discussed. Authors need to provide scientific reasons for their results and not just citations of other works.

Conclusion: Needs more adjustments.

Comments on the Quality of English Language

The quality of the English can still be improved. 

Author Response

Comments and Suggestions for Authors: 

  1. Title: Change title to “Innovation and utilization of Functional Feed Additives from Maize By-products in broiler chickens”
  • We agreed with the suggestion of the reviewer. Thus, we have revised the title (Line 2-3 of revised manuscript with Track Changes).

  1. Abstract: Poorly written. Needs improvement. Contains more of introductory statements than the results of the study.
  • Thank you for your feedback. I have revised the abstract to better reflect the results of the study, reducing the introductory statements. I hope you find the updated version more to your satisfaction. Please let me know if there are any further adjustments needed.
  • Thus, we have revised introductory (Line 28-29 of revised manuscript) and the results (Line 36-43 of revised manuscript with Track Changes).

  1. Line 20 -21: Rephrase to “The objectives of this study were to evaluate the impact of feeding varying levels of SMS on growth performance, carcass characteristics, meat quality, cecal microbiota, and small intestinal morphology in broilers.”
  • We agreed with the suggestion of the reviewer. Thus, we have revised the objectives in simple summary
  • The objectives of this study were to evaluate the impact of feeding varying levels of SMS on growth performance, blood biochemicals, carcass characteristics, meat quality, cecal microbiota, and small intestinal morphology in broilers.” (Line 22-25 of revised manuscript with Track Changes).

  1. Line 34-35: Rephrase. First line of result should not read “however”. Was it significant? Statement should read “…broilers in CON group had decreased average daily feed intake compared to ……”. Include other important significant parameters….
  • We agreed with the suggestion of the reviewer. Thus, we have revised the result as shown (Line 37-39 of revised manuscript with Track Changes).
  • We have added other important significant parameters and revised more correct follow.

      Final body weight and average daily gain in broilers fed AGP diet the higher than broilers fed CON, SMS0.5, and SMS1.0 diets. (Line 36-37 of revised manuscript with Track Changes)

      The addition of AGP and SMS2.0 diets were improved feed conversion ratio and reduced feed cost per gain in broilers. In broilers fed CON diet the highest serum cholesterol while AGP diet increased triglyceride. (Line 39-41 of revised manuscript with Track Changes)

      Dietary supplementation of SMS improved some carcass characteristic and ceacum microbial in broilers, especially SMS2.0 diet. (Line 41-42 of revised manuscript with Track Changes)

      Broiler fed CON and SMS0.5 showed worsened villus height/crypt depth ration of duodenum histology. (Line 42-43 revised manuscript with Track Changes)

  1. Line 35-37: Check the conclusion part.
  • We checked with the suggestion of the reviewer. Thus, we have revised the conclusion (Line 43-45 of revised manuscript with Track Changes).

  1. Line 41-43: Provide Reference
  • Thanks for the suggestions. Thus, we have revised and added reference as shown (Line 49-50 of revised manuscript with Track Changes).

FYI:

  • Charoenratana, S.; Anukul, C.; Rosset, P.M. Food Sovereignty and Food Security: Livelihood Strategies Pursued by Farmers during the Maize Monoculture Boom in Northern Thailand. Sustainability 2021, 13, 9821

  1. Line 51-55: while the emphasis on enzymes. Delete.
  • We agreed with the suggestion of the reviewer. Thus, we have delete the emphasis on enzymes (Line 61-65 of revised manuscript with Track Changes).

  1. Line 57: change “improve to “improved”
  • We agreed with the suggestion of the reviewer. Thus, we have revised “improves to “improved” (Line 67 of revised manuscript with Track Changes).

  1. Line 65: remove “medicinal”
  • We agreed with the suggestion of the reviewer. Thus, we have removed as shown (Line 75 of revised manuscript with Track Changes).

  1. Line 70-71: Provide more references.
  • We agreed with the suggestion of the reviewer. Thus, we have added reference (Line 83 of revised manuscript).

FYI:

  • Jing, P.; Zhao, S.-J.; Lu, M.-M.; Cai, Z.; Pang, J.; Song, L.-H. Multiple-fingerprint analysis for investigating quality control of Flammulina velutipes fruiting body polysaccharides. J. Agric. Food Chem. 2014, 62, 12128-12133.
  • Zhang, Z.; Lv, G.; He, W.; Shi, L.; Pan, H.; Fan, L. Effects of extraction methods on the antioxidant activities of polysaccharides obtained from Flammulina velutipes. Carbohydr. Polym. 2013, 98, 1524-1531.

  1. Line 81: Remove “were”
  • We agreed with the suggestion of the reviewer. Thus, we have removed “were” (Line 91 of revised manuscript with Track Changes).

  1. Line 82: Use “Ross”
  • We agreed with the suggestion of the reviewer. Thus, we have revised (Line 92 of revised manuscript with Track Changes).

  1. Line 84-85: Rewrite as “consisted of amoxicillin 100 g kg-1 and colistin 400×106 IU kg-1 at 0.25 g kg-1 ; Otamix a.c., Octa Memorial Co., Ltd., Bangkok, Thailand)
  • We agreed with the suggestion of the reviewer. Thus, we have revised (Line 94-97 of revised manuscript with Track Changes).

  1. Line 95: Stick with one “Mortality or mortality rate”
  • We agreed with the suggestion of the reviewer. Thus, we have revised (Line 110 of revised manuscript with Track Changes).

  1. Line 98: Change “broiler” to “bird”
  • We agreed with the suggestion of the reviewer. Thus, we have changed “broiler” to “bird” (Line 113 of revised manuscript with Track Changes).

  1. Line 120-124: Authors should rephrase this section of the procedure to provide more clarity. Not clear “The meat sample was absorbed moisture…….
  • We agreed with the suggestion of the reviewer. Thus, we have revised (Line 136-139 of revised manuscript with Track Changes).

  1. Line 127: Provide details of the “mechanical device”.
  • We agreed with the suggestion of the reviewer. Thus, we have provided the details of mechanical device (Line 144 of revised manuscript with Track Changes).

  1. Line 131: Remove “briefly”. Should read One gram (1 g)
  • We agreed with the suggestion of the reviewer. Thus, we have revised (Line 147 of revised manuscript with Track Changes).

  1. Line 132: Rephrase this “9-ml 0.1% peptone solution combined with a vortex mixer.”
  • We agreed with the suggestion of the reviewer. Thus, we have revised (Line 148-149 of revised manuscript with Track Changes).

  1. Line 133: To what dilution factor? Provide the precise composition of the MRS and EMB plates.
  • We agreed with the suggestion of the reviewer. Thus, we have revised (Line 149-152 of revised manuscript with Track Changes).

  1. Line 144: Rewrite as “Subsequently, the tissue specimens underwent a standardized three step processing protocol,”
  • Thank you for your input. I have rewritten the sentence as requested: Subsequently, the tissue specimens underwent a standardized three-step processing protocol." Please review and let me know if there are any further adjustments needed. (Line 161-162 of revised manuscript with Track Changes).

  1. Line 145: Remove “which included three primary steps:”
  • We agreed with the suggestion of the reviewer. Thus, we have removed (Line 162-163 of revised manuscript with Track Changes).

  1. Line 146: Provide model of the rotary microtome
  • We agreed with the suggestion of the reviewer. Thus, we have provided the model of rotary microtome as shown (Line 164 of revised manuscript with Track Changes).

  1. Line 155: Author should run linear regressions to model and analyse the relationships between the variables assessed and SMS supplementation
  • Thank you for your suggestion, We have conducted both linear and quadratic regressions to examine the relationships between the assessed variables and SMS supplementation. (Line 178-179 of revised manuscript with Track Changes).

  1. Line 168: Delete “In contrast, the inclusion of SMS at a concentration of 2.0 g/kg (SMS2.0) did not significantly affect ADFI (p > 0.05).” Not correct.
  • We agreed with the suggestion of the reviewer. Thus, we have deleted (Line 188-189 of revised manuscript with Track Changes).

  1. Line 167: Did authors use p<0.01???? Adjust this.
  • We think that the suggestion of the reviewer refer Line 170. We agreed with the suggestion of the reviewer. Thus, we have revised (Line 190 of revised manuscript with Track Changes).

  1. Line 173: Statement not correct. AGP, SMS1.0 and SMS2.0 had significantly lowered Serum Cholesterol content. Check your Table.
  • Due to AGP had the highest triglyceride, thus, we have rewritten by “SMS1.0 and SMS2.0 had significantly lowered serum cholesterol content” as shown (Line 195-197 of revised manuscript with Track Changes).

  1. Line 191: Start on a new line or paragraph.
  • We agreed with the reviewer. Thus, we have started the new line in Line 208 (Line 221 of revised manuscript).

  1. Line 197: Authors should note that healthy chickens have beneficial coli in their intestines. Is the E. coli in the study beneficial or pathogenic as assumed by the Authors?
  • A very good point, we recognize that healthy chickens can host beneficial strains of E. coli as part of their normal gut microbiota. However, in this study, we focused on E. coli as a pathogenic organism that can lead to adverse health effects.

  1. Line 210-212: Delete the line.
  • We agreed with the reviewer. Thus, we have deleted (Line 243-245 of revised manuscript with Track Changes).

  1. Line 215-217 Delete.
  • We agreed with the reviewer. Thus, we have deleted (Line 248-250 of revised manuscript with Track Changes).

  1. Line 220: No details on Energy of diets in “Table 1. Ingredients and chemical composition of the experimental diets.”
  • We agreed with the reviewer. Therefore, we have included detailed information on the energy content of diets in Table 1. (Line 255 of revised manuscript with Track Changes).

  1. Line 227-229: Inconsistency in assigning of superscripts. E.g. FCR, the lowest had the first alphabet unlike the other parameters
  • We agreed with the reviewer. Thus, we have revised the superscripts of FCR (Line 267 of revised manuscript with Track Changes).

  1. Line 254: Check values of the redness. It is suspect.
  • Due to the typing error, thus, we have corrected the redness (Line 294 of revised manuscript with Track Changes).
  1. Line 277-280: Figure 1 does not show anything. Not necessary. Authors did not use any annotations.
  • Thank you for your feedback. We have removed Figure 1 as requested. Please review and let us know if any further adjustments are needed. (Lines 320-324 of revised manuscript with Track Changes).

  1. Line 284-285: What is the link to the enzyme??
  • SMS as by-products from mushroom cultivation which have residual fungal mycelium as source of enzymes.

  1. Line 297-318: Authors only cited works and did not provide scientific reasons/justifications for their results. Authors should review more on the composition of SMS and link its bioactive components in relation to some of the performance indices. Too much of unnecessary citations.
  • We have revised and provided scientific reasons for these results as shown (Line 362-373 of revised manuscript with Track Changes).

  1. Line 320: What is the implication of the reduction in the HDL-Cholesterol? Authors failed to provide reasons for the reduction in the cholesterol composition of the serum. Authors can read and check: Ekunseitan, D. A., Ekunseitan, O. F., Odutayo, O. J., & Adeyemi, P. T. (2017). Pleurotus Ostreatus: Its Effect on Carcass, Serum Metabolites and Meat Lipoprotein Content of Broiler Chickens. Pertanika Journal of Tropical Agricultural Science, 40(4): 629 – 638 for help. There more examples.
  • We have provided the additional reasons (Line 377-391 of revised manuscript with Track Changes).

  1. Line 330: Authors should check Table 3. Delete “cholesterol, or triglyceride levels:
  • We have checked the content by modify the sentence (Line 398-401 of revised manuscript with Track Changes).

  1. Line 342-345: Delete the line.
  • We agreed with the reviewer. Thus, we have deleted the line (Lines 411-414 of revised manuscript with Track Changes).

  1. Line 353-362: Delete the lines.
  • We agreed with the reviewer. Thus, we have deleted the line (Lines 428-437 of revised manuscript with Track Changes).

  1. Line 378-394: Too much citations. Authors has not properly explained the reasons for the changes in the morphological changes in the duodenum and the jejunum and it’s implication on the birds.
  • We agreed with the reviewer. Thus, we have revised the additional reasons (Lines 462-470 and 486-490 of revised manuscript with Track Changes).

  1. Line 394: How is this achieved? Explain this.
  • We have revised the additional reasons (Lines 486-490 of revised manuscript with Track Changes).

  1. Results: Still more corrections
  • Thank you for your feedback. I have reviewed and made the necessary corrections. I hope the revised version meets your expectations. Please let me know if there are any further adjustments needed.

  1. Discussion: Not properly discussed. Authors need to provide scientific reasons for their results and not just citations of other works.
  • Discussion: Not properly discussed. Authors need to provide scientific reasons for their results and not just citations of other works.

  1. Conclusion: Needs more adjustments.
  • Thank you for your input. I've made further adjustments to the conclusion for better clarity and completeness. Please review the revised version and let me know if there are any more areas that need refinement. (Lines 495-502 of revised manuscript with Track Changes).

Comments on the Quality of English Language:

The quality of the English can still be improved.

  • Thank you for your feedback. We have made additional revisions to enhance the quality of the English. Please review the updated version and let me know if further adjustments are required.

Reviewer 2 Report

Comments and Suggestions for Authors

General comments

This study investigates the addition of spent mushroom substrate (SMS) on different performance, blood, carcass and microbial parameters in broilers. Although, the study was well executed and a lot of results are produced, there is a lack of compositional analysis of the SMS (feed additive). This information on the composition of the SMS is however crucial to clarify and explain the reported results. Which compounds present in the SMS contribute to the changes in the measured parameters? SMS is also described as an exogenous enzyme product, but no enzyme activities were measured. Next to this, the effect of dosage of SMS is not discussed although it entails valuable information. The discussion is rather tedious to read as there are no explanations or correlations given for the reported effects, it remains very descriptive, while readers want answers the questions as: why did SMS cause these effects? Which compounds are responsible for these observed effects etc.?

Specific comments

Abstract

L33: add the AGP used in the abstract

Materials & Methods

L92: a justification for the weekly 4-hour fasting period is needed. What was the purpose of this fasting period in the experimental design?

L99: revise the unit used for centrifugation

L130: How representative are caecal digesta when collected after a fasting period? Can it not be that some microbial species will already be starved and hence an underestimation of this composition can occur.

L131: caecal fluid – better refer to caecal digesta

L155: Regarding the statistical analysis, you did not analyse the effect of dosage, although it might be interesting as you used 3 different dosages for the SMS. I recommend to also add this analysis to the manuscript. In addition, was the one-way ANOVA used for all measured parameters?

Results

L168: This result is incorrect in the table. Each value is reported as significant different, even a difference of 0.2 g/d, which is very unlikely given the reported SEM. Revise this result in Table 2 (L227)

L170: It might be informative to report on the mortality rates in each group.

L172: For the serum biochemical parameters it might be of interest to perform analysis for the effect of dosage. As it seems that 2.0 g/kg is less optimal compared to 1.0 g/kg regarding the blood parameters. A discussion on the dose effect throughout the results and discussion part of the manuscript is therefore recommended.

L220: indicate what the % means in the chemical composition of the diets: as fed, dm base? Chemical composition of the SMS is missing in this part. This information is however crucial to explain the observed effects. Now the authors are speculating on which compounds can possibly be responsible for the observed effects. Highly recommended to analyse the chemical composition and the enzyme activities of the SMS.

L245 + L254: The authors might want to report the insignificant data in the tables as supplementary data. There are already quite some tables in this paper. Reporting on insignificant data is relevant, but maybe not all of these should be included in the body of the manuscript.

Discussion

Overall the discussion part should be revised thoroughly. The focus of the discussion remains very descriptive; mainly describing the results of other studies without really focussing on similarities or dissimilarities, and more importantly the causalities of the observed effects. This should be improved throughout the whole discussion part.

L284: The SMS is reported as exogenous enzyme product, but there is no information given on the enzymes that are present in this substrate. An overview and measurement of the relevant enzymes present should be added.

L292-294: This sentence does not read very well.

L294-318: An overload of references is given here. Try to relate your findings to relevant references; discuss similarities and dissimilarities and explain. Explain also why improved performance effects such as FCR are observed when adding SMS. The causal effects are not discussed.

L346-362: What is the point you want to make. Here it remains very descriptive. Did you expect adverse or positive effects on meat quality due to the addition of SMS, and why? Not clear at the moment.

L363: Increase? You measured only 3 bacterial groups, hence it is incorrect to indicate this as an overall increase of the microbial population.

L371: Define which polysaccharides are present

L372: Do you refer here to the microbiota or the mushrooms for the VFA production. Presence of VFA in the SMS? Not clear.

L395-397: which compounds do cause the antioxidant properties of SMS? A better clarification for the relation between microbiota and gut morphology is also needed.

Conclusion

L400: You are stating that 2.0 g/kg might be the best dosage, but you didn’t discuss or analyse the dosage effects more in-depth. Should be added in the manuscript to make this conclusive remark.

L402: Be careful with saying that it can effectively replace the AGPs. You only measured a couple of parameters in this study (selective bacterial groups on plates, no immunomodulating properties, …). Therefore it is recommended to formulate this sentence with more care, as in there is potential for SMS to act as an alternative for AGP.

Comments on the Quality of English Language

Only minor editing of some sentences is required to improve the understanding and readibility of it. No major remarks on the quality of English.

Author Response

General comments:

  1. This study investigates the addition of spent mushroom substrate (SMS) on different performance, blood, carcass and microbial parameters in broilers. Although, the study was well executed and a lot of results are produced, there is a lack of compositional analysis of the SMS (feed additive). This information on the composition of the SMS is however crucial to clarify and explain the reported results. Which compounds present in the SMS contribute to the changes in the measured parameters? SMS is also described as an exogenous enzyme product, but no enzyme activities were measured. Next to this, the effect of dosage of SMS is not discussed although it entails valuable information. The discussion is rather tedious to read as there are no explanations or correlations given for the reported effects, it remains very descriptive, while readers want answers the questions as: why did SMS cause these effects? Which compounds are responsible for these observed effects etc.?
  • We have added the composition of the SMS as shown Table 2. (Lines 262 of revised manuscript with Track Changes).
  • Compounds of SMS including antioxidant but did not show in this paper.
  • Utilization dosage of SMS at 2.0 g/kg in diet can improve performance, serum cholesterol content, some carcass characteristic, cecal microbiota, and small intestinal histology of broilers.
  • The discussion, we did rephrase the explanations or correlation for SMS cause these effects.

Specific comments

Abstract:

2. L33: add the AGP used in the abstract

  • We apologize for not being able to add the details you requested. However, this information is already provided in the materials and methods section (2.1 Animals and Experiment Design, Lines 94-97 of revised manuscript). Adding it to the abstract would exceed the 200-word limit.

Materials and Methods: 

3. L92: a justification for the weekly 4-hour fasting period is needed. What was the purpose of this fasting period in the experimental design?

  • We have revised the details of fasting methods used (Lines 105-108 of revised manuscript with Track Changes).

4. L99: revise the unit used for centrifugation

  • We agreed with the reviewer. Thus, we have revised the unit used for centrifugation as shown (Lines 114 of revised manuscript with Track Changes).

5. L130: How representative are caecal digesta when collected after a fasting period? Can it not be that some microbial species will already be starved and hence an underestimation of this composition can occur.

  • Caecal digesta collected after a fasting period may not fully represent the microbial community as it exists under normal feeding conditions. Fasting can lead to changes in microbial activity and composition, as certain species may respond differently to the lack of nutrients. While some microbes are resilient to short-term fasting, others may be more sensitive, potentially affecting the overall microbial profile captured in the sample.
  • Yes, it is possible that some microbial species may be starved during fasting, leading to an underestimation of their presence in the caecal digesta. However, we did quickly analysis the microbial count after collecting sample for protection loss the microbial.

 6. L131: caecal fluid – better refer to caecal digesta

  • We agreed with the reviewer. Thus, we have revise “caecal fluid” to “caecal digesta” as shown (Line 147 of revised manuscript with Track Changes).

7. L155: Regarding the statistical analysis, you did not analyse the effect of dosage, although it might be interesting as you used 3 different dosages for the SMS. I recommend to also add this analysis to the manuscript. In addition, was the one-way ANOVA used for all measured parameters?

  • Indeed, this study utilized one-way ANOVA for all measured parameters, as indicated in lines 174-177 of the revised manuscript with track changes.
  • We have conducted both linear and quadratic regressions to examine the relationships between the assessed variables and SMS supplementation. (Line 178-179 of revised manuscript with Track Changes).

Results:

8. L168: This result is incorrect in the table. Each value is reported as significant different, even a difference of 0.2 g/d, which is very unlikely given the reported SEM. Revise this result in Table 2 (L227)

  • We agreed with the reviewer. Thus, we have revised the results (Line 267 of revised manuscript with Track Changes).

9. L170: It might be informative to report on the mortality rates in each group.

  • We agreed with the reviewer. Thus, we have removed the mortality rates.

10. L172: For the serum biochemical parameters it might be of interest to perform analysis for the effect of dosage. As it seems that 2.0 g/kg is less optimal compared to 1.0 g/kg regarding the blood parameters. A discussion on the dose effect throughout the results and discussion part of the manuscript is therefore recommended.

  • We did improve the contents by modifying the sentences and make the new sentences (Lines 195-197 of revised manuscript with Track Changes) and provide reason (Lines 392-407 of revised manuscript with Track Changes) as shown.

11. L220: indicate what the % means in the chemical composition of the diets: as fed, dm base? Chemical composition of the SMS is missing in this part. This information is however crucial to explain the observed effects. Now the authors are speculating on which compounds can possibly be responsible for the observed effects. Highly recommended to analyse the chemical composition and the enzyme activities of the SMS.

  • The chemical composition of the diet are based on as fed
  • We have added the chemical composition of SMS as shown Table 2 (Lines 262 of revised manuscript with Track Changes).
  • We have added the enzyme activity of SMS as shown Figure 1 (Lines 264-266 of revised manuscript with Track Changes).

12. L245 + L254: The authors might want to report the insignificant data in the tables as supplementary data. There are already quite some tables in this paper. Reporting on insignificant data is relevant, but maybe not all of these should be included in the body of the manuscript.

  • We have revised the data to exclude the supplementary information.

Discussion:

 13. Overall the discussion part should be revised thoroughly. The focus of the discussion remains very descriptive; mainly describing the results of other studies without really focussing on similarities or dissimilarities, and more importantly the causalities of the observed effects. This should be improved throughout the whole discussion part.

  • We have improved the discussion part by modify more correct and provide reason about effect of SMS on each parameters.

14. L284: The SMS is reported as exogenous enzyme product, but there is no information given on the enzymes that are present in this substrate. An overview and measurement of the relevant enzymes present should be added. SMS

  • We have added the enzyme activity of SMS in Figure as shown (Lines 264-266 of revised manuscript with Track Changes).

15. L292-294: This sentence does not read very well.

  • We have revised the sentences (Lines 336-338 of revised manuscript with Track Changes).

16. L294-318: An overload of references is given here. Try to relate your findings to relevant references; discuss similarities and dissimilarities and explain. Explain also why improved performance effects such as FCR are observed when adding SMS. The causal effects are not discussed.

  • We have improved the contents by modifying the sentences and revised the sentences (Lines 362-373 of revised manuscript with Track Changes).

17. L346-362: What is the point you want to make. Here it remains very descriptive. Did you expect adverse or positive effects on meat quality due to the addition of SMS, and why? Not clear at the moment.

  • We have described about result and provided the reasons (Lines 437-440 of revised manuscript with Track Changes).
  • We expected that dietary supplementation with SMS may improve carcass characteristics and meat quality, but SMS affected only some parameters of carcass.

18. L363: Increase? You measured only 3 bacterial groups, hence it is incorrect to indicate this as an overall increase of the microbial population.

  • We have improved the contents by modifying the sentences and make the new sentences as shown (Lines 441 of revised manuscript with Track Changes).

19. L371: Define which polysaccharides are present.

  • We have revised the details of polysaccharides (Lines 449-451 of revised manuscript with Track Changes).

20. L372: Do you refer here to the microbiota or the mushrooms for the VFA production. Presence of VFA in the SMS? Not clear.

  • We have revised the sentences to refer the VFA production by microbiota (Lines 449-453 of revised manuscript with Track Changes).

21. L395-397: which compounds do cause the antioxidant properties of SMS? A better clarification for the relation between microbiota and gut morphology is also needed.

  • We have revised the sentences to clarify the compounds responsible for the antioxidant properties and the relationship between microbiota and gut morphology (Lines 486-490 of revised manuscript with Track Changes).

Discussion:

22. L400: You are stating that 2.0 g/kg might be the best dosage, but you didn’t discuss or analyse the dosage effects more in-depth. Should be added in the manuscript to make this conclusive remark.

  • We have conducted both linear and quadratic regressions to examine the dosage effects of SMS supplementation. (Line 178-179 of revised manuscript with Track Changes).

23. L402: Be careful with saying that it can effectively replace the AGPs. You only measured a couple of parameters in this study (selective bacterial groups on plates, no immunomodulating properties, …). Therefore it is recommended to formulate this sentence with more care, as in there is potential for SMS to act as an alternative for AGP.

  • We agreed with the suggestion of the reviewer. Thus, we have revised the conclusion to indicate the potential for SMS to act as an alternative for AGP. (Lines 501-502 of revised manuscript with Track Changes).

Comments on the Quality of English Language:

24. Only minor editing of some sentences is required to improve the understanding and readibility of it. No major remarks on the quality of English.

  • We have reviewed and enhanced the understanding and readability of the sentences, resulting in more accurate and improved phrasing.

Reviewer 3 Report

Comments and Suggestions for Authors

The article entitled "Innovation of Functional Feed Additives from Maize By-products for “Change Corncob to Exogenous Enzyme Product” "is well-structured and highly engaging. However, the research's objective could be more clearly emphasized in the summary.

The introduction is well organized, providing a concise overview that effectively piques the reader's curiosity about the core research.

The methods are adequately described, offering sufficient detail for replication.

The results are clearly presented, and the discussion is robust, drawing on relevant studies to support the findings.

However, the conclusions are brief, and given the breadth of results presented earlier, they risk being overlooked in the broader context of the research. Expanding the conclusion to better synthesize key findings would strengthen the overall impact. For instance, the results regarding the small intestinal histology are quite interesting and weren't highlighted in the conclusion part. Additionally, the cecal microbiota in broilers receiving the control diet (CON), antibiotic growth promoter (AGP), and spent mushroom substrate (SMS) supplementation revealed significant results, particularly with an increase in Lactobacillus observed in the SMS-supplemented groups.

Author Response

Comments and Suggestions for Authors:

1. The article entitled "Innovation of Functional Feed Additives from Maize By-products for “Change Corncob to Exogenous Enzyme Product” "is well-structured and highly engaging. However, the research's objective could be more clearly emphasized in the summary.

  • We agreed with the suggestion of the reviewer. Thus, we have revised the objectives in simple summary

2. The introduction is well organized, providing a concise overview that effectively piques the reader's curiosity about the core research.

  • Thank you for your kind words. We're glad to hear that the introduction successfully provides a concise overview and piques the reader's curiosity. If there are any other aspects that need refinement or improvement, please let us know.

3. The methods are adequately described, offering sufficient detail for replication.

  • Thank you for your feedback. We are pleased to know that the methods section is considered adequately detailed for replication.

4. The results are clearly presented, and the discussion is robust, drawing on relevant studies to support the findings.

  • We appreciate the reviewer’s feedback. We are glad to hear that the results and discussion sections are well received.

5. However, the conclusions are brief, and given the breadth of results presented earlier, they risk being overlooked in the broader context of the research. Expanding the conclusion to better synthesize key findings would strengthen the overall impact. For instance, the results regarding the small intestinal histology are quite interesting and weren't highlighted in the conclusion part. Additionally, the cecal microbiota in broilers receiving the control diet (CON), antibiotic growth promoter (AGP), and spent mushroom substrate (SMS) supplementation revealed significant results, particularly with an increase in Lactobacillus observed in the SMS-supplemented groups.

  • We have improved the contents by modifying the conclusion (Lines 495-502 of revised manuscript with Track Changes).

Round 2

Reviewer 1 Report

Comments and Suggestions for Authors

Line 136-137: Revise to "The meat sample was gently wiped with tissue paper to remove the surface water, weight was recorded and then placed in a heat-resistant bag and sealed securely."

Author Response

Comments and Suggestions for Authors:

Line 136-137: Revise to "The meat sample was gently wiped with tissue paper to remove the surface water, weight was recorded and then placed in a heat-resistant bag and sealed securely."

  • We agreed with the suggestion of the reviewer. Thus, we have revised “The meat sample was absorbed moisture by gently wipe with tissue paper to remove the surface water and recorded as weight. Place the sampled in a heat-resistant bag and sealed securely” to "The meat sample was gently wiped with tissue paper to remove the surface water, weight was recorded and then placed in a heat-resistant bag and sealed securely," as shown (Lines 138-141 of revised manuscript with Track Changes).

Reviewer 2 Report

Comments and Suggestions for Authors

Abstract

L28-45: Please edit the English language in the reviewed lines in the abstract. Incorrect usage of English language in the corrected parts.

L33-35: Incorrect English language in the sentence describing the treatments. Rewrite. “were added/supplemented to the diet) ‘receiving to diet’ is not a correct expression

 Materials & Methods

L105-108: Why was this info deleted? I am confused now, was a fasting period part of the experiment or not? You did not give clear arguments in your rebuttal and manuscript why you initially used a fasting period. Be clear about this and do not hide info from the readers by just deleting some info on the experimental set-up.

L111: English language should be improved in this sentence. ‘trail’ ‘were’ are all incorrect.

L114: rpm = rotations per minute, the per min in this sentence is hence redundant information

L178 + L197: Revise English language

 Results

L191;L206; L218; L224; L234; L252: Just reporting on “contrasts were significant” is not of added value to the reader. These sentences are not meaningful as such. You should describe it in function of your obtained result; for example: A quadratic trend in … which means that a higher supplementation of SMS reduced the response of … Please pay again attention to English language. ‘some vale’ ‘some value’ does not give an indication on what values were changed. Revise thoroughly.

Table 2; L262: Only report DM or moisture content, these parameters give the same information. In addition, only 50% of the composition is explained by the parameters presented in the table. If it is not protein, fat or fibre, what else is in the SMS substrate that can contribute to the observations? Not reporting on 50% of the DM content of the SMS substrate is incorrect. Revise the chemical composition of the substrate you have added.

L266: ‘gastrointestinal in vivo’. Incorrect English expression. What do you mean? Measured at the level of the small intestines in vivo?

L321: Figure 1 (or 2 in this case) on the morphology does not have a caption anymore. Deleted in the revised version. Every figure should have a clear description and title in the caption. Please include a correct caption for this figure.

Original comment 12 (L245 + 254 on reducing the amount of data in the tables by adding some info into supplementary data) was misunderstood. Authors have excluded the supplementary data and removed this from the manuscript, while this was not the intention of this feedback. Please revise this comment again.

Discussion & Conclusion

Overall, the effect of supplementation level is not well described in the discussion part, despite the fact that the authors did add an extra statistical analysis in the results part. Try to highlight the most important outcomes throughout your discussion.

L363: The compounds mentioned in this paragraph are not presented in the compositional analysis. Still 50% of your substrate is unknown.

L379: Revise English language

L441: “improve in caecal microbial population”. I suppose you mean the microbial composition? Be more clear.

L451; L488: Revise English language

L496: ‘enhance caecal microbial population’. Specify what you mean : enhancement of VFA production or shift in microbial composition. Now this sentence does not provide a lot of info.

Comments on the Quality of English Language

English language used in the revised parts of the manuscript (as well as in the answers in the rebuttal letter) need moderate editing. The English language usage is of a lower quality in these parts compared to the first version of the manuscript.

Author Response

Comments and Suggestions for Authors: 

Abstract:

L28-45: Please edit the English language in the reviewed lines in the abstract. Incorrect usage of English language in the corrected parts.

  • We agreed with the suggestion of the reviewer. Thus, we have revised more correct in the abstract as shown (Lines 28-46 of revised manuscript).

L33-35: Incorrect English language in the sentence describing the treatments. Rewrite. “were added/supplemented to the diet) ‘receiving to diet’ is not a correct expression

  • We agreed with the suggestion of the reviewer. Thus, we have revised more correct in the abstract as shown (Lines 33-36 of revised manuscript).

Materials and Methods:

L105-108: Why was this info deleted? I am confused now, was a fasting period part of the experiment or not? You did not give clear arguments in your rebuttal and manuscript why you initially used a fasting period. Be clear about this and do not hide info from the readers by just deleting some info on the experimental set-up.

  • We apologize for the confusion. There was no fasting period mentioned in the growth performance section; it was a typing error. We have decided to remove that sentence.

L111: English language should be improved in this sentence. ‘trail’ ‘were’ are all incorrect.

  • We agreed with the reviewer. Thus, we have revised “were” to “was” as shown (Lines 112 of revised manuscript with Track Changes).

L114: rpm = rotations per minute, the per min in this sentence is hence redundant information

  • We agreed with the reviewer. Thus, we have revised “rpm” to “rotations per minute” as shown (Lines 116-117 of revised manuscript with Track Changes).

L178 + L197: Revise English language

  • We agreed with the reviewer. Thus, we have revised the sentence. (Line 181 and Line 201 of revised manuscript with Track Changes).

Results:

L191; L206; L218; L224; L234; L252: Just reporting on “contrasts were significant” is not of added value to the reader. These sentences are not meaningful as such. You should describe it in function of your obtained result; for example: A quadratic trend in … which means that a higher supplementation of SMS reduced the response of … Please pay again attention to English language. ‘some vale’ ‘some value’ does not give an indication on what values were changed. Revise thoroughly.

  • We agreed with the reviewer. Thus, we have revised the results (Line 194-197, Line 210-213, Line 221-222, Line 225-227, Line 232-233, and Line 242-244, and Line 261-263 of revised manuscript with Track Changes, respectively).

Table 2; L262: Only report DM or moisture content, these parameters give the same information. In addition, only 50% of the composition is explained by the parameters presented in the table. If it is not protein, fat or fibre, what else is in the SMS substrate that can contribute to the observations? Not reporting on 50% of the DM content of the SMS substrate is incorrect. Revise the chemical composition of the substrate you have added.

  • We agreed with the reviewer. Thus, we have removed moisture content.
  • We have revised chemical composition of SMS as shown in Table 2 (Line 272 of revised manuscript).

L266: ‘gastrointestinal in vivo’. Incorrect English expression. What do you mean? Measured at the level of the small intestines in vivo?

  • We have revised the sentences (Lines 275 of revised manuscript with Track Changes).

L321: Figure 1 (or 2 in this case) on the morphology does not have a caption anymore. Deleted in the revised version. Every figure should have a clear description and title in the caption. Please include a correct caption for this figure.

  • We have deleted Figure 1 (or 2 in this case) due to receiving the suggestion of the reviewer that Figure 1 (or 2 in this case) does not show anything and not necessary in round 1.

Original comment 12 (L245 + 254 on reducing the amount of data in the tables by adding some info into supplementary data) was misunderstood. Authors have excluded the supplementary data and removed this from the manuscript, while this was not the intention of this feedback. Please revise this comment again.

  • Thank you for your valuable feedback. We appreciate your suggestion regarding the reporting of insignificant data. We have revised to include the insignificant results in supplementary table S1 and S2 while maintaining a concise presentation of significant findings in the main text (Lines 215-233 of revised manuscript).

Discussion and Conclusion:

Overall, the effect of supplementation level is not well described in the discussion part, despite the fact that the authors did add an extra statistical analysis in the results part. Try to highlight the most important outcomes throughout your discussion.

  • Thank you for your feedback. We recognize the need to better articulate the effects of supplementation levels in the discussion section. We have revised to clearly highlight the most significant outcomes, emphasizing how varying levels of supplementation influence the results. (Lines 382-385 and 451-455 of revised manuscript with Track Changes).

L363: The compounds mentioned in this paragraph are not presented in the compositional analysis. Still 50% of your substrate is unknown.

  • Thank you for your valuable suggestion about the compositional analysis in our study. The observation that 50% of the substrate is unidentified is indeed significant. We have incorporated a discussion on the potential implications of these unidentified compounds for our overall findings. (Lines 458-460 of revised manuscript with Track Changes).

L379: Revise English language

  • We have revised the new sentences as shown (Lines 382-385 of revised manuscript with Track Changes).

L441: “improve in caecal microbial population”. I suppose you mean the microbial composition? Be more clear.

  • We have revised the sentence to describe the effects on microbial composition (Lines 446 of revised manuscript with Track Changes).

L451; L488: Revise English language

  • We have revised these sentences (Lines 460 and Line 497 of revised manuscript with Track Changes).

L496: ‘enhance caecal microbial population’. Specify what you mean : enhancement of VFA production or shift in microbial composition. Now this sentence does not provide a lot of info.

  • We have revised the sentence to clarify the effect on the composition of cecal microbiota. (Lines 504-505 of revised manuscript with Track Changes).

Comments on the Quality of English Language:

English language used in the revised parts of the manuscript (as well as in the answers in the rebuttal letter) need moderate editing. The English language usage is of a lower quality in these parts compared to the first version of the manuscript.

  • Thank you for your feedback. We appreciate your observations regarding the English language in the revised sections. We have made the necessary edits to enhance clarity and ensure consistency with the quality of the original manuscript.
